# Expression of Serpin Family E Member 1 (SERPINE1) Is Associated with Poor Prognosis of Gastric Adenocarcinoma

**DOI:** 10.3390/biomedicines11123346

**Published:** 2023-12-18

**Authors:** Jie Lv, Chunyang Yu, Hanhan Tian, Tao Li, Changhua Yu

**Affiliations:** 1Department of Radiotherapy, The Affiliated Huaian No. 1 People’s Hospital of Nanjing Medical University, Huanghe West Road, Huaiyin District, Huai’an 223300, China; mr.lvjie@163.com (J.L.); tianhanshandong@163.com (H.T.); lt605166@163.com (T.L.); 2Department of Cardiology, The Affiliated Huaian No. 1 People’s Hospital of Nanjing Medical University, Huai’an 223300, China; hayyycy@njmu.edu.cn

**Keywords:** gastric adenocarcinoma, SERPINE1, The Cancer Genome Atlas, prognosis

## Abstract

Background: The aberrant expression of serpin family E member 1 (SERPINE1) is associated with carcinogenesis. This study assessed the alteration of SERPINE1 expression for an association with gastric adenocarcinoma prognosis. Methods: The Cancer Genome Atlas (TCGA) dataset was applied to investigate the impact of SERPINE1 expression on the survival of patients afflicted with gastric cancer. Subsequently, 136 samples from the Affiliated Huaian No. 1 People’s Hospital of Nanjing Medical University were subjected to qRT-PCR and Western blot to validate the expression level of SERPINE1 between tumor and adjacent normal tissues. The correlation between the expression of SERPINE1 with the clinicopathological features in TCGA patients was analyzed using Wilcoxon signed-rank and logistic regression tests. The potential molecular mechanism associated with SERPINE1 expression in gastric cancer were confirmed using gene set enrichment analysis (GSEA). Results: The TCGA data showed that SERPINE1 was overexpressed in tumor tissues compared to normal mucosae and associated with the tumor T stage and pathological grade. SERPINE1 overexpression was associated with the poor overall survival (OS) of patients. The findings were confirmed with 136 patients, that is, SERPINE1 expression was associated with poor OS (hazard ratio (HR): 1.82; 95% confidence interval (CI): 0.84–1.83; *p* = 0.012)) as an independent predictor (HR: 2.11, 95% CI: 0.81–2.34; *p* = 0.009). The resulting data were further processed by GSEA showed that SERPINE1 overexpression was associated with the activation of EPITHELIAL_MESENCHYMAL_TRANSITION, TNFA_SIGNALING_VIA_NFKB, INFLAMMATORY_RESPONSE, ANGIOGENESIS, and HYPOXIA. Conclusions: SERPINE1 overexpression is associated with a poor gastric cancer prognosis.

## 1. Introduction

Gastric adenocarcinoma, commonly known as gastric cancer, is a common malignancy with a heavy burden and is still a significant global health problem. Owing to its oftentimes advanced stage at diagnosis, the mortality of patients afflicted with gastric cancer is high, making it the fifth for incidence and fourth for mortality worldwide [1], with 769,000 deaths globally in 2020. It is noteworthy that over 40% of new cases and deaths of gastric cancer occur in China, with only 27.4% of patients achieving five-year survival [2]. Besides, the detection of early gastric cancer is still challenging, as clinical symptoms frequently occur late during gastric cancer progression, thus lessening alternatives for surgical treatment [3]. Even with the ongoing therapeutic options, the optimal treatment for an individual patient with gastric cancer is tough to determine because of the large heterogeneity in patients [4]. Thus, further research on novel biomarkers and a better understanding of their molecular mechanisms are the key to developing novel strategies to improve the personal prognosis and treatment of gastric cancer.

Tumor metastasis as well as resistance to chemotherapy have been identified as the main causes of death among patients with gastric adenocarcinoma [5]. However, the illustration of the molecular mechanism related to the development and metastasis of gastric cancer remains limited. It is well accepted that the occurrence and progression of gastric cancer is attributed to the perturbations of the transcriptome in response to epigenetic and genetic alterations [6]. Thus, genetic alterations have been increasingly identified as potential targets for gastric cancer. On the basement of this, we aimed to explore effective therapeutic targets, for the purpose of providing a promising biomarker for gastric cancer treatment. Serpin family E member 1 (SERPINE1), also known as plasminogen activator inhibitor-1 (PAI-1), however, cannot only inhibit the plasminogen activator but also plays a paradoxical role in tumorigenesis [7]. SERPINE1 has multiple pro-tumor roles in tumorigenesis by promoting tumor angiogenesis [8], preventing excessive proteolysis, and maintaining extracellular matrix integrity [9]. Additionally, SERPINE1 exerts protection of tumor cells from chemotherapy-induced apoptosis via suppressing intracellular caspase 3 [10]. In gastric cancer, several previous studies demonstrated that SERPINE1 was overexpressed in tumor tissues compared to the level in normal gastric mucosae, and SERPINE1 expression was associated with a poor prognosis for patients [11,12,13]. SERPINE1 is one of the ten hub genes involved in the pathogenesis and prognosis of gastric cancer [14]. Indeed, another previous study showed that SERPINE1 played a role in maintaining gastric mucosal organization in hypergastrinemia [15]. However, a previous study indicated that the level of SERPINE1 expression was high in both gastric cancer and corresponding normal tissues [16].

In this study, we assessed the prognostic value of SERPINE1 expression in human gastric cancer using The Cancer Genome Atlas (TCGA) data and confirmed the results using our own cohort of patients. We then performed GSEA to identify the potentially related biological pathways regulated by SERPINE1 in gastric cancer development.

## 2. Materials and Methods

### 2.1. TCGA Database and Data Retrieval

Data on SERPINE1 expression in human gastric adenoma or cancer was searched in the TCGA database and retrieved from the website (https://portal.gdc.cancer.gov/, accessed on 1 May 2023). The clinicopathological and survival data for patients were also retrieved from the TCGA website. The expression difference in the discrete variables was previously visualized; therefore, we retrieved the SERPINE1 expression data for 406 gastric cancer cases for analysis. We next performed differential gene expression analysis in the TCGA-STAD dataset, which was split into high-and low-SERPINE1 groups according to the median SERPINE1 transcript per million (TPM).

Pan-cancer and the Genotype-Tissue Expression (GTEx) transcriptome data and overall survival data were downloaded from the University of California, Santa Cruz (UCSC) Xena database (http://xena.ucsc.edu, accessed on 1 May 2023), and all data were normalized by the Sangerbox (http://sangerbox.com, accessed on 1 May 2023) online tool to analyze the differences in SERPINE1 expression between cancer and normal tissues. Similarly, to investigate the prognosis of SERPINE1 in human cancers, based on The Cancer Genome Atlas (TCGA) database, we conducted univariate Cox proportional risk regression models using the Sangerbox (http://sangerbox.com, accessed on 1 May 2023) online network to explore the association between SERPINE1 and overall survival (OS) in different cancers.

### 2.2. Our Patients and Tissue Samples

In this study, we recruited a total of 136 patients who were histologically diagnosed with gastric cancer at the Affiliated Huaian No. 1 People’s Hospital of Nanjing Medical University between January 2012 and June 2013 and underwent D2 lymph node dissection and radical gastrectomy. These patients did not receive preoperative chemotherapy and/or radiotherapy. Gastric cancer was diagnosed by two experienced pathologists according to the eighth edition of the International Union Against Cancer (UICC) guidelines [17]. After recruitment, we collected tumor and matched normal tissue samples from each patient, submerged the tissue samples in RNA later solutions (Ambion, Life Technologies, Thermo-Fisher Scientific, Waltham, MA, USA) at 4 °C and stored them at −80 °C until use. This study was approved by the Ethics Committee of the Affiliated Huaian No.1 People’s Hospital of Nanjing Medical University, and informed consent was obtained from each patient before participation in this study.

### 2.3. Quantitative Reverse Transcriptase-Polymerase Chain Reaction (qRT-PCR)

Total RNA was isolated from 136 tumor and normal tissue samples using the TRIzol reagent (Invitrogen, Carlsbad, CA, USA) and reverse-transcribed into cDNA using the First-strand cDNA synthesis kit for the Prime Script RT Master Mix (TaKaRa Biotechnology Co., Ltd., Dalian, China) according to the manufacturers instructions. Quantitative PCR was performed using the standard SYBR Green PCR kit (Thermo, Waltham, MA, USA) at conditions of 95 °C for 2 min and 40 cycles of 95 °C for 15 s and 60 °C for 60 s. The qPCR primers were SERPINE1, 5′-GCCCGATGGCCATTACTACGACATCCTG-3′ and 5-GGAAAGGCAACATGACC-3′; glyceraldehyde 3-phosphate dehydrogenase (GAPDH), 5′-AAGGTCATCCCTGAGCTGAA-3′ and 5′-TGACAAAGTGGTCGTTGAGG-3′. The GAPDH mRNA was used as an endogenous control, and the level of SERPINE1 expression was quantified using the ΔΔCt method. The experiment was performed in triplicate and repeated at least once.

### 2.4. Western Blot

Total cellular protein was extracted from the tissue samples using the radioimmunoprecipitation assay (RIPA) lysis buffer (Abcam, Cambridge, UK). The protein concentration was measured using the bicinchoninic acid (BCA) protein assay kit (Pierce, Rockford, IL, USA). Afterward, the protein samples were separated in 10% or 12% sodium dodecyl sulfate-polyacrylamide gel electrophoresis (SDS-PAGE) gels and transferred onto nitrocellulose membranes. For Western blotting, the membranes were first blocked in 10% skimmed milk solution in phosphate buffered saline (PBS)-Tween 20 (PBS-T) at room temperature for 1 h and then incubated with the primary antibody at 4 °C overnight. The membranes were subsequently incubated with secondary antibodies conjugated with horseradish peroxidase at room temperature for 2 h and detected using the enhanced chemiluminescence (ECL) kit (Tanon, Shanghai, China). The anti-SERPINE1 antibody was purchased from Millipore (Billerica, MA, USA), while the anti-GAPDH antibody was obtained from Sigma Chemicals (St. Louis, MO, USA).

### 2.5. Statistical Analysis

The TCGA data were statistically analyzed using R (v.3.6.0) software (https://cran.r-project.org/bin/windows/base/old/3.6.0/, accessed on 1 May 2023). The paired *t*-test was performed to compare SERPINE1 expression in tumor and normal tissues, while the Wilcoxon signed-rank and logistic regression tests were used to analyze the SERPINE1 expression to determine the association with the clinicopathological characteristics of the patients. Kaplan-Meier curves and the log-rank test were used to analyze the OS stratified by SERPINE1 expression in the gastric cancer patients. Univariate and multivariate Cox regression analyses were also performed to assess the clinicopathological characteristics of the patients and SERPINE1 expression to predict the patients’ OS. All *p*-values were two-sided, and a *p*-value < 0.05 was considered statistically significant.

To explore the relationship between SERPINE1 expression and cancer hallmarks, we downloaded 50 cancer hallmarks from the MSIGDB database (https://www.gsea-msigdb.org/gsea/msigdb, accessed on 1 May 2023) for differential and correlation analysis. Besides, we performed GSEA [18] analysis (v2.0) to identify SERPINE1-associated biological pathways based on differential expression and correlation analysis. In brief, we ranked the related genes according to their degree of association with the expression of SERPINE1 after selecting the default parameters and the significance threshold through 1000 permutations and analyses. Next, we calculated the false discovery rate (FDR); when the FDR reached 0.25, we considered this gene set to be significantly enriched.

## 3. Results

### 3.1. Characteristics of the TCGA Patients

In this study, we retrieved the data for 406 gastric cancer cases from the TCGA database for SERPINE1 expression analysis. The median age of the patients was 65.7 years old (Table 1) with 63.1% men and 36.9% women. The histological grades of the tumors were classified as well-, moderately, and poorly differentiated (2.5%, 37.3%, and 60.2%, respectively), and the tumors were classified as stage I in 56 patients (14.7%), stage II in 118 (31.1%), stage III in 167 (43.9%), and stage IV in 39 (10.3%). Two hundred and sixty-seven (68.6%) of 389 patients had lymph node metastasis, and 27 (7%) of 388 patients had distant tumor metastasis.

### 3.2. Expression Level Analysis and Prognosis Association of SERPINE1 in Pan-Cancer Patient

We analyzed the expression of SERPINE1 mRNA in the TCGA database and found that SERPINE1 was high in 12 types of cancer, including GBM, COAD, COAD-READ, BRCA, ESCA, STES, STAD, HNSC, KIRC, LUSC, THCA, and READ (Figure 1A). Besides, due to the limited normal sample number in the TCGA database, we integrated the data of TCGA with GTEx databases to assess expression of SERPINE1 in pan-cancer types. The results showed that SERPINE1 was differentially expressed in 15 cancers (Figure 1B).

Based on the TCGA database, we created forest plots for univariate COX regression analysis using Sangerbox to investigate the prognostic significance of SERPINE1 expression in various cancers. OS analysis showed that high SERPINE1expression was associated with shorter OS in GBM, LGG, KIPAN, STAD, UVM, MESO, STES, HNSC, CESC, LIHC, LUSC, LUAD, KIRP, PAAD, BLCA, KIRC, and THCA (*p*  <  0.05, hr > 1, Figure 1C).

### 3.3. Expression and Association of SERPINE1 Level with Clinicopathological Variables in TCGA-STAD Patients

We analyzed the SERPINE1 expression in a total of 406 samples with gastric cancer from the TCGA dataset. Our data showed that SERPINE1 expression was higher in tumor tissues than in normal tissues (Figure 2A,B), and that SERPINE1 expression was associated with pathologic stage (Figure 2C) and pathological grade (Figure 2D). SERPINE1 expression level at the T1, T2, T3, and T4 stage was significantly higher than normal (Figure 2E). However, there is no difference in N and M compared to normal (Figure 2F,G) Besides, there is no difference in age and gender compared to the control (Figure 2H,I). Patients with high SERPINE1 expression level had worse overall survival (Figure 2J). A receiver operating characteristic (ROC) curve and the area under the curve (AUC) of SERPINE1 in STAD showed good predictive ability (Figure 2K). Based on logistic regression analysis, we separated the gastric cancer samples into SERPINE1 high and low expression groups using the median level as the cut-off value. We found that high SERPINE1 expression was associated with poor prognostic clinicopathological features, such as a poorly differentiated histological grade (odds ratio (OR) = 1.84 (0.44–9.17) versus (vs.) a well-differentiated grade, T3 vs. T1 T stage (OR = 2.22 (0.82–7.52)) and T4 vs. T1 (OR = 3.56 (0.91–9.86; Table 2)).

### 3.4. Association of SERPINE1 Expression with Survival in House Gastric Cancer Patients

We next confirmed our findings regarding SERPINE1 expression with a cohort of 136 patients using qRT-PCR and Western blot. In particular, the expression level of SERPINE1 was higher in gastric cancer tissues than in normal tissues in our cohort (Figure 3A,B).

We plotted the Kaplan–Meier survival curves stratified by high vs. low level of SERPINE1 expression and performed a log-rank test. Our data showed that in both the 406 TCGA cases and our own data comprising 136 cases, high SERPINE1 expression was associated with worse OS and disease-free survival (DFS) for patients (all *p* < 0.05; Figure 3C,D). Univariate analysis clearly showed an inverse relationship between poor OS and SERPINE1 expression with a hazard ratio (HR) of 1.82 and 95% confidence interval (CI) of 0.84–1.83 (*p* = 0.012; Table 3). Other clinicopathological variables, such as lymph node metastasis, T stage, and TNM stage, were also associated with a poor prognosis for patients (Table 3). Multivariate analysis showed that only gender and SERPINE1 expression predicted the survival of these patients, with an HR of 2.11 (95% CI: 0.81–2.34, *p* = 0.009; Table 3).

### 3.5. Association of SERPINE1 Expression and Tumor Hallmarks Based on Differential Expression and Correlation Analysis

To explore the relationship between SERPINE1 expression and gastric cancer, we conducted differential expression and correlation analysis to identify possible SERPINE1-related hallmark pathway involved in the regulation of gastric cancer progression and metastasis. As shown in Figure 4A, we found that the most significantly hallmark pathways, including MYC_TARGETS_V1, EPITHELIAL_MESENCHYMAL_TRANSITION, OXIDATIVE_PHOSPHORYLATION, and TNFA_SIGNALING_VIA_NFKB, were upregulated in gastric cancer samples with high SERPINE1 expression. Furthermore, EPITHELIAL_MESENCHYMAL_TRANSITION, TGF_BETA_SIGNALING, HYPOXIA, and ANGIOGENESIS were highly associated with SERPINE1 expression (Figure 4B). Specifically, ANGIOGENESIS and EPITHELIAL_MESENCHYMAL_TRANSITION remain the highest associate with SERPINE1 expression (Spearman correlation coefficient > 0.6, *p*-value < 0.05, Figure 4C,D).

### 3.6. GSEA for Association of SERPINE1 Expression and Tumor Hallmarks

Although our current data are descriptive, we performed a bioinformatic analysis to identify possible SERPINE1-related signaling pathways involved in the regulation of gastric cancer progression and metastasis. We adopt two strategies to explore the correlation between SERPINE1 expression and tumor hallmarks, including differential expression analysis (Figure 5A) and guilt of association (Figure 5B). We found that the most significantly enriched signaling pathways in gastric cancer samples with high SERPINE1 expression according to the normalized enrichment score (NES) were pathways related to cancer, specifically, the EPITHELIAL_MESENCHYMAL_TRANSITION, TNFA_SIGNALING_VIA_NFKB, INFLAMMATORY_RESPONSE, KRAS_SIGNALING_UP, and ALLOGRAFT_REJECTION (*p* < 0.05, Figure 5A). Similarly, genes related to SERPINE1 expression were significantly enriched in EPITHELIAL_MESENCHYMAL_TRANSITION, TNFA_SIGNALING_VIA_NFKB, INFLAMMATORY_RESPONSE, ANGIOGENESIS, and HYPOXIA (*p* < 0.05, Figure 5B).

## 4. Discussion

Gastric cancer is still a commonly diagnosed malignant tumor globally, especially in Asian countries [19]. With the advancement of diagnostic radiology, most early gastric cancer can be detected by esophagogastroduodenoscopy, as well as barium X-ray with photofluorography, and a 5-year survival rate for gastric cancer can be achieved >95% with the help of surgery treatment. Unfortunately, more than 70% of patients are diagnosed during the middle or advanced stage, so the optimal time of surgery is most probably missed [20,21]. The increasing incidence and mortality of gastric cancer place a considerable burden on the social economy, which urgently needs an accurate prognostic evaluation as well as early diagnosis biomarkers.

SERPINE1 is the major controller of the uPA system, which plays a key role in tumor cell migration and metastasis [22]. Aberrant SERPINE1 expression and the association with a poor prognosis for gastric cancer are well documented in the PubMed literature [23,24]. However, discrepant results do exist, showing a similarly high expression of SERPINE1 in both gastric cancer and corresponding normal tissues [16]. Based on this, we decided to thoroughly investigate the role of SERPINE1 in gastric cancer patients from the TCGA dataset and validate in our own cohort to provide a reference for individualized treatment.

In this present study, we first detected the differential expression of SERPINE1 between the tumor and normal tissues of gastric cancer patients from the TCGA dataset and found that SERPINE1 was the upregulated genes in tumor tissues. Subsequently, we extracted RNA and protein from gastric tumor tissues and adjacent samples and verified the expression level of SERPINE1 by qRT-PCR test and Western Blot. The results coincided with previous studies [25,26] and further confirmed that (1) SERPINE1 mRNA or protein was highly expressed in gastric cancer tissues vs. the normal mucosae from both the TCGA dataset (n = 406) and our own cohort (n = 136), revealing that SERPINE1 may serve as a promising diagnostic biomarker; (2) High expression level of SERPINE1 was significantly associated with advanced tumor stage and pathology grade (*p* < 0.05), suggesting that upregulation of SERPINE1 notably promoted the development of gastric cancer; and (3) Overexpression of SERPINE1 expression contributed to a poor disease-free survival and OS for patients with gastric cancer (*p* < 0.05), indicating the potential applicability of SERPINE1 being a prognostic biomarker for gastric cancer patients. In depth, GSEA analysis revealed that the high SERPINE1 expression was significantly enriched in EPITHELIAL_MESENCHYMAL_TRANSITION, TNFA_SIGNALING_VIA_NFKB, INFLAMMATORY_RESPONSE, ANGIOGENESIS, and HYPOXIA in gastric cancer. These results further revealed that SERPINE1 could be a promising therapeutic target and prognostic indicator for gastric cancer.

SERPINE1 is a member of the Serpinprotease inhibitor (SERPIN) superfamily. SERPINE1 mainly regulates the plasminogen activator system, and it possesses an activating effect on tissue-type plasminogen and an inhibitory effect on urokinase-type plasminogen [27], which is not conducive to the conversion of plasminogen into active protease plasmin. The SERPINE1 gene, which encode glycoprotein of approximately 50 kD, is localized at chromosome 7q21.2-q22 and possesses several polymorphisms in its promoter region for different transcription processes [28]. SERPINE1 expression is thought to be regulated by various transcriptional factors, like HIF-1 [29], p53 [30], and TGF-β1 [31], as well as epigenetically [32] and by miR-145 [33]. It has been reported that SERPINE1 regulates plasminogen activators and urokinase which transform the pro-enzyme plasminogen to plasmin, and thus facilitates cellular invasion by degradation of the extracellular matrix and activation of matrix metalloproteinases [34]. Previous studies revealed that *Helicobacter pylori* infection induced the expression of the urokinase plasminogen activator system in gastric epithelial cells [35] as well as SERPINE1 expression in gastric cancer cells [36], whereas SERPINE1 RNAi suppressed gastric cancer metastasis [37]. These data further support the role of SERPINE1 in the development of gastric cancer and the benefits of SERPINE1 expression knockdown in the control of gastric cancer progression.

However, data on SERPINE1 are inconsistent; for example, SERPINE1 expression may not be associated with any clinicopathological features in squamous cell carcinomas [38]. Our current data support the significant association of SERPINE1 expression with the T4 stage and M1 of gastric cancer and a poor prognosis as an independent predictor. Molecularly, SERPINE1 expression-related TGF-β signaling could transform human keratinocytes or normal cells through an increase in cell invasion or migration [39]. In breast cancer and fibrosarcoma cells, SERPINE1 overexpression through the activation of the PI3K-Akt pathway induced tumor cell migration capacity and invasion ability [40]. Moreover, the low-density lipoprotein receptor-related protein 1 (LRP1) was able to interact with the migration-promoting effects of SERPINE1 to influence the activity of the JAK/STAT pathway [41]. Our current bioinformatic analysis data further support the notion of SERPINE1′s involvement in the progression of cancer, including gastric cancer.

Significantly, from the SERPINE1-related GSEA, it can be seen that TNFA_SIGNALING_VIA_NFKB and INFLAMMATORY_RESPONSE is significantly related to SERPINE1, which implies that SERPINE1 is related to immune signaling pathway. In addition, some published articles have reported significant correlation between SERPINE1 and immunity factor in pan-cancer [42]. cGAS-cGAMP-STING, the three musketeers of cytosolic DNA sensing and signaling [43], initialize the immune process. In future research, we will explore the relationship between SERPINE1 and immune recognition, further elucidating the relationship between SERPINE1 immune promotion and inhibition in gastric cancer.

Our current study is merely a descriptive and associative investigation of SERPINE1 expression to determine the association with gastric cancer progression and prognosis, which has led to some limitations, such as calculating the cut-off point and high vs. low SERPINE1 expression. In conclusion, our current study demonstrated and confirmed that the overexpression of SERPINE1 in gastric cancer patients was associated with a poor prognosis and advanced clinical characteristics. Moreover, EPITHELIAL_MESENCHYMAL_TRANSITION, TNFA_SIGNALING_VIA_NFKB, INFLAMMATORY_RESPONSE, ANGIOGENESIS, and HYPOXIA may be the signaling pathways regulated by SERPINE1 in gastric cancer. A further experimental validation of SERPINE1 functions in gastric cells will help researchers better understand the role of SERPINE1 in gastric cancer development and progression and assess whether targeting SERPINE1 expression can be used as a novel approach to control gastric cancer. Further study is needed to uncover the underlying molecular mechanism of action.

## Figures and Tables

**Figure 1 biomedicines-11-03346-f001:**
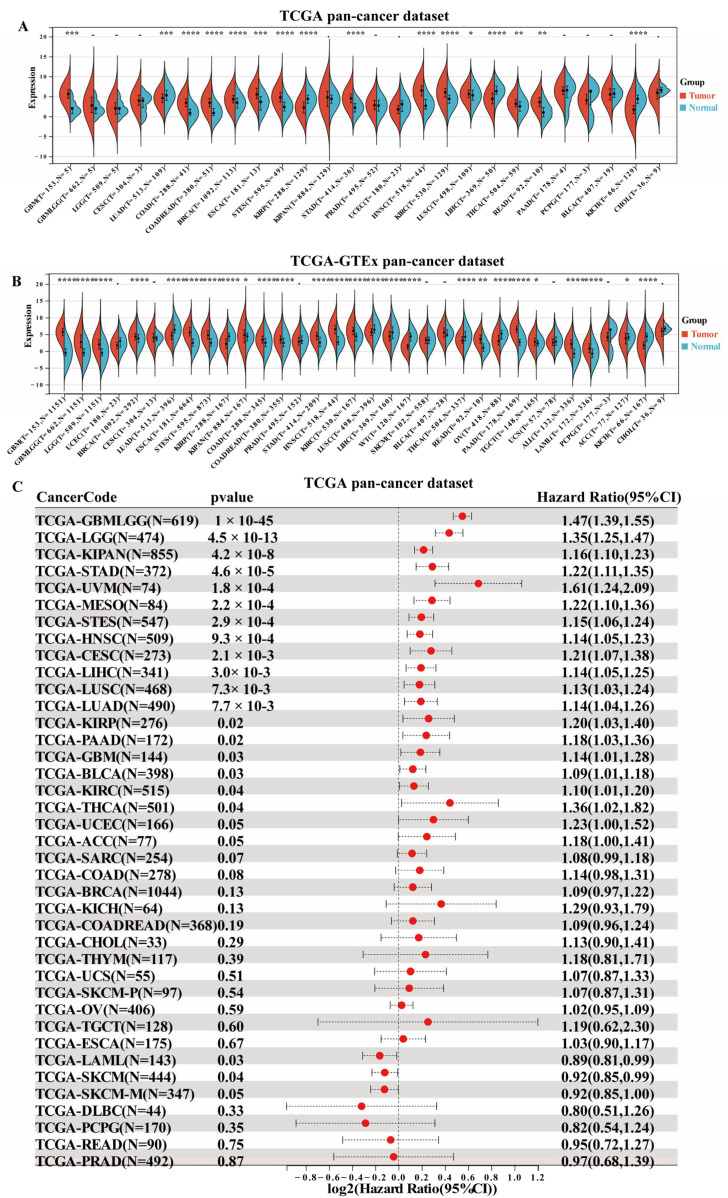
Expression and overall survival association of SERPINE1 in pan-cancer. (**A**) Expression analysis of SERPINE1 mRNA from pan-cancer and normal tissues. (**B**) Combining TCGA and GTEx databases to obtain SERPINE1 mRNA expression levels. (**C**) Prognosis (overall survival) analysis of SERPINE1 in various cancers from the TCGA database using the Sangerbox website tool. “*” represents *p* < 0.05, “**” represents *p* < 0.01, “***” represents *p* < 0.001, “****” represents *p* < 0.0001.

**Figure 2 biomedicines-11-03346-f002:**
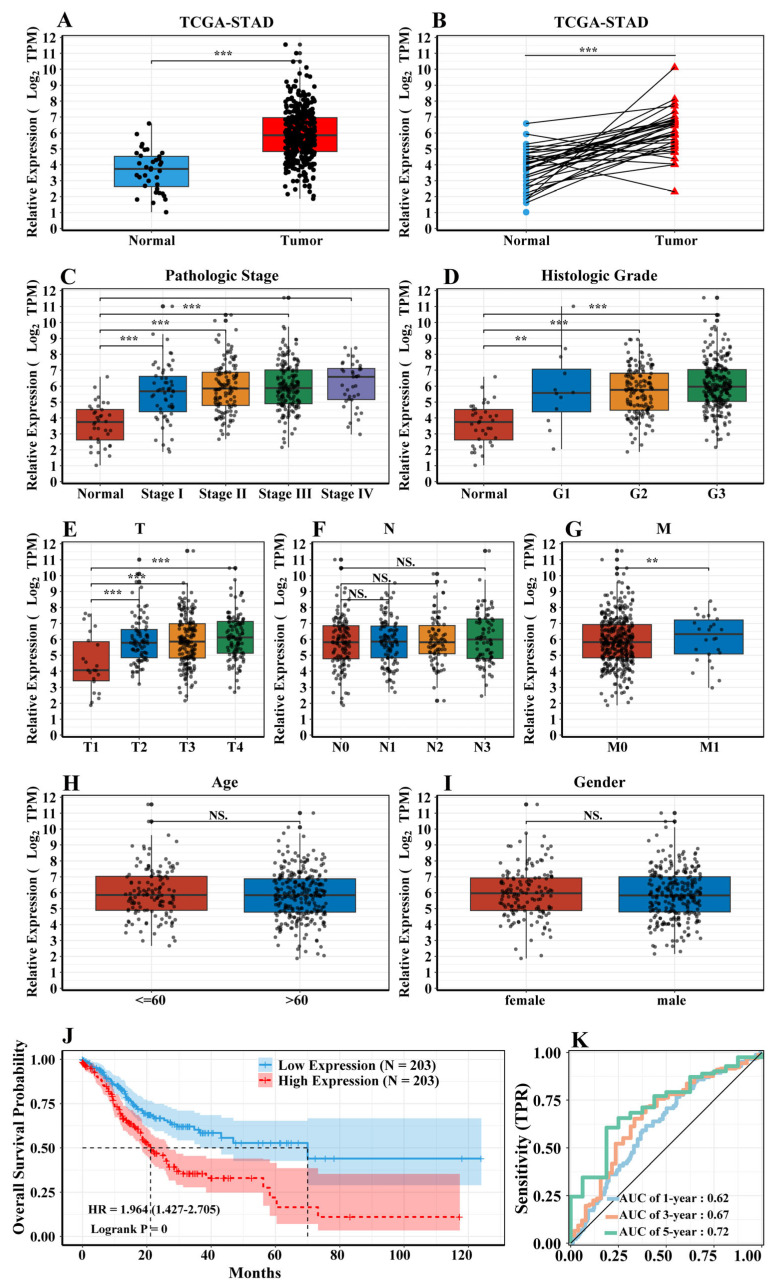
Expression and association of SERPINE1 with clinicopathological characteristics in the TCGA-STAD dataset. (**A**) SERPINE1 expression in STAD tissues and normal tissues (Wilcoxon rank sum test). (**B**) SERPINE1 expression in STAD tissues and adjacent noncancerous tissues (Wilcoxon signed-rank test). (**C**) Expression level of SERPINE1 in patients with different pathological stages (Kruskal–Wallis test). (**D**) Expression level of SERPINE1 in patients with different histologic grades (Kruskal–Wallis test). (**E**–**G**) Expression level of SERPINE1 in patients with TNM (Kruskal–Wallis test). (**H**) Expression level of SERPINE1 in patients with age (Wilcoxon rank sum test). (**I**) Expression level of SERPINE1 in patients with gender (Wilcoxon rank sum test). (**J**) Overall survivals of patients with high and low SERPINE1 expression (log-rank test). (**K**) A receiver operating characteristic (ROC) curve and the area under the curve (AUC) of SERPINE1 at 1 year, 3 years, and 5 years. “**” represents *p* < 0.01, “***” represents *p* < 0.001, “NS” represents *p* > 0.05.

**Figure 3 biomedicines-11-03346-f003:**
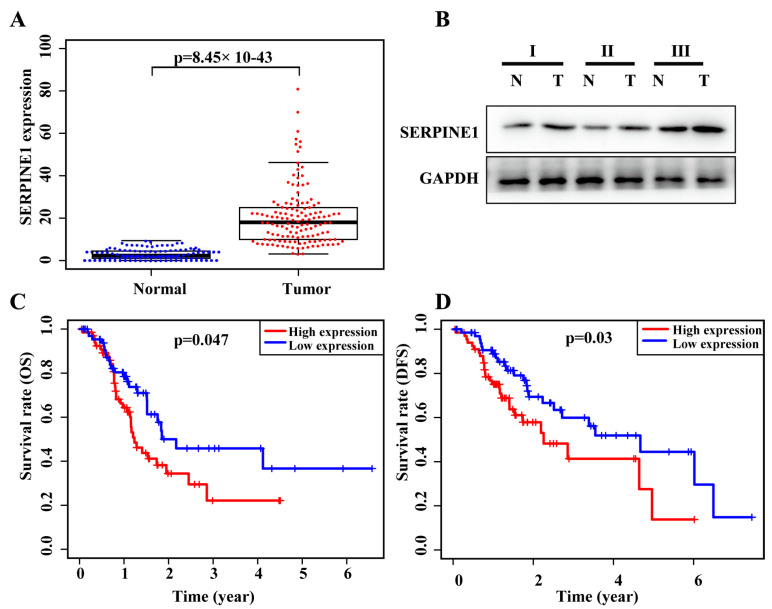
Expression and association of SERPINE1 in house cohort of 136 gastric cancer tissues. (**A**) SERPINE1 mRNA levels were assessed in 136 primary gastric cancer and normal tissues using qRT-PCR. (**B**) SERPINE1 protein levels were analyzed in normal and gastric cancer tissues using Western blotting. (**C**) The association of SERPINE1 expression with overall survival (OS) among 136 gastric cancer patients was assessed using Kaplan-Meier curves and a log-rank test. (**D**) The association of SERPINE1 expression with the disease-free survival (DFS) of 136 gastric cancer patients was assessed using Kaplan–Meier curves and a log-rank test.

**Figure 4 biomedicines-11-03346-f004:**
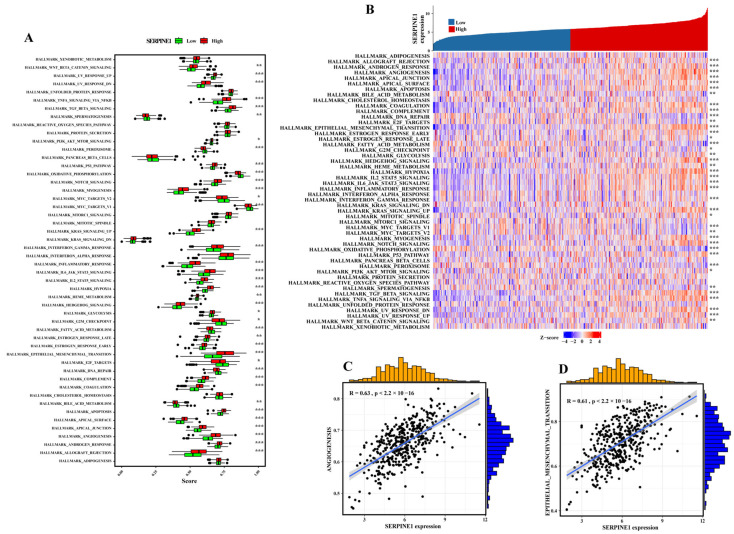
Association of SERPINE1 expression and tumor hallmarks based on differential expression and correlation analysis. (**A**) Differential 50 cancer hallmark score between high- and low- SERPINE1 groups. (**B**) The correlation between SERPINE1 expression and 50 cancer hallmark score. (**C**,**D**) The most prominent cancer hallmark associated with the differential co-expression of SERPINE1 expression (Spearman correlation coefficient > 0.6, *p*-value < 0.05). “*” represents *p* < 0.05, “**” represents *p* < 0.01, “***” represents *p* < 0.001.

**Figure 5 biomedicines-11-03346-f005:**
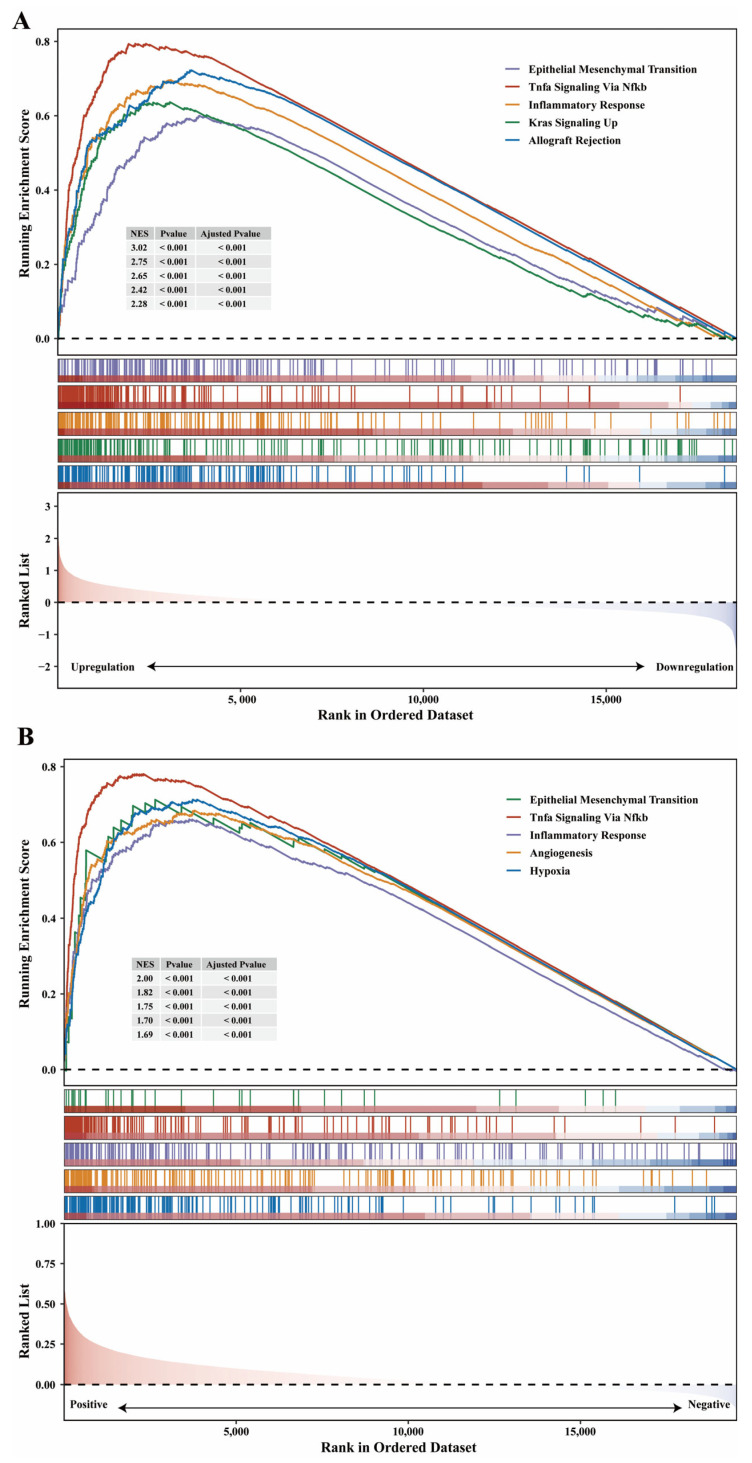
Identification of SERPINE1-associated gene pathways using gene set enrichment analysis (GSEA). (**A**) Gene set enrichment analysis for patients classified into SERPINE1 high group and SERPINE1 low group according to the median value of SERPINE1 expression. (**B**) Via a guilt by association approach, we performed GSEA to explore the potential cancer hallmark association of SERPINE1 in stomach cancer. NES, normalized enrichment score; NOM-*p*-value, nominal *p*-value.

**Table 1 biomedicines-11-03346-t001:** The clinicopathological variables of TCGA patients with gastric cancer.

Variables	Total (*n* = 406)	%
Age at diagnosis	65.7 (30–90)	
Gender		
Male	256	63.1
Female	150	36.9
Histological Grade		
Well-differentiated	10	2.5
Moderately differentiated	149	37.3
Poorly differentiated	240	60.2
TNM Stage		
I	56	14.7
II	118	31.1
III	167	43.9
IV	39	10.3
Tumor stage		
T1	23	5.8
T2	85	21.5
T3	185	46.7
T4	103	26
Node positivity		
N0	122	31.4
N1	109	28
N2	80	20.6
N3	78	20
Distant metastasis		
Positive	361	93
Negative	27	7

**Table 2 biomedicines-11-03346-t002:** Association of SERPINE1 expression with clinicopathologic variables for TCGA patients.

Variables	*n*	Odds Ratio in SERPINE1 Expression	*p* Value
Age (continuous)	342	1.31 (0.86–2.01)	0.21
Gender (male vs. female)	344	1.02 (0.66–1.58)	0.94
Histological Grade (well- vs. poorly differentiated)	337	1.84 (0.44–9.17)	0.042
TNM Stage			
II vs. I	152	1.40 (1.72–2.87)	0.29
III vs. I	185	1.44 (0.73–2.72)	0.15
IV vs. I	85	1.75 (0.72–4.27)	0.064
T			
T2 vs. T1	94	2.05 (0.72–6.37)	0.081
T3 vs. T1	176	2.22 (0.82–7.52)	0.042
T4 vs. T1	104	3.56 (0.91–9.86)	0.004
N (positive vs. negative)	328	1.11 (0.63–1.96)	0.72
Distant metastasis (positive vs. negative)	322	1.31 (0.64–2.72)	0.45

**Table 3 biomedicines-11-03346-t003:** Univariate and multivariate analyses of clinicopathological variables and SERPINE1 expression for prediction of OS of TCGA patients.

Variables	Univariate Analysis				Multivariate Analysis			
	HR	HR.95L	HR.95H	*p* Value	HR	HR.95L	HR.95H	*p* Value
Age at diagnosis	1.02	1.01	1.05	0.004	1.04	1.02	1.07	7.85 × 10^−5^
Gender	1.49	0.97	2.29	0.062	1.58	1.01	2.47	0.043
Histologic grade	1.25	0.85	1.84	0.25	1.28	0.85	1.93	0.22
TNM Stage	1.51	1.19	1.91	0.001	1.33	0.84	2.11	0.21
T	1.27	0.99	1.63	0.05	1.12	0.79	1.58	0.52
N	2.06	1.07	3.98	0.03	2.21	0.93	5.28	0.074
Distant metastasis	1.25	1.05	1.49	0.013	1.05	0.8	1.38	0.72
SERPINE1	1.82	0.84	1.83	0.012	2.11	0.81	2.34	0.009

## Data Availability

The current data are available upon reasonable request.

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
