# Peer review of "Expression of Serpin Family E Member 1 (SERPINE1) Is Associated with Poor Prognosis of Gastric Adenocarcinoma"

_biomedicines, 2023, doi:10.3390/biomedicines11123346_

Round 1

Reviewer 1 Report

Comments and Suggestions for Authors

The aim of this study was to investigate the association between the expression of Serpin family E member 1 (SERPINE1) and the prognosis of gastric adenocarcinoma. The study utilized data from The Cancer Genome Atlas (TCGA) to examine how SERPINE1 expression affects the survival of gastric cancer patients. Additionally, the study aimed to validate these findings by analyzing 136 samples from patients at the Affiliated Huaian No.1 People's Hospital of Nanjing Medical University using qRT-PCR and Western blot. The authors also sought to explore the correlation between SERPINE1 expression and various clinicopathological features in TCGA patients. Finally, the study aimed to uncover potential molecular mechanisms associated with SERPINE1 expression in gastric cancer through Gene Set Enrichment Analysis (GSEA). The results showed that SERPINE1 was overexpressed in tumor tissues compared to normal mucosae, and this overexpression was associated with poor overall survival of gastric cancer patients. GSEA analysis revealed that SERPINE1 overexpression was linked to the activation of cancer-related signaling pathways, including the transforming growth factor beta (TGF-β), Janus Tyrosine Kinase-Signal Transducer and Activator of Transcription (JAK/STAT), and Wnt pathways. In conclusion, the study demonstrated that SERPINE1 overexpression is associated with a poor prognosis in gastric cancer, and it provided insights into the potential molecular mechanisms underlying this association.
Q1- Can the authors explain the distribution of age among the 406 gastric cancer patients in this study and its potential implications for the research findings?

Q2- What is the significance of the observed gender distribution in this study, and does it impact the results related to SERPINE1 expression?

Q3- How do the histological grades of the tumors correlate with SERPINE1 expression, and what might this imply about gastric cancer progression?

Q4- Among the TCGA patients, can the authors elaborate on the distribution of TNM stages and how they relate to SERPINE1 expression?

Q5- Could the authors provide further details on the relationship between T stage and SERPINE1 expression in gastric cancer patients?

Q6- What are the implications of the findings that SERPINE1 expression is higher in tumor tissues compared to normal tissues? How does this relate to the study's objectives?

Q7- Can the authors explain how the logistic regression analysis was used to categorize gastric cancer samples into SERPINE1-high and low expression groups, and what clinical significance does this hold?

Q8- In the multivariate analysis, what is the significance of gender and SERPINE1 expression as predictors of survival, and how do they compare to other clinicopathological variables?

Q9- Could the authors provide more details on the bioinformatic analysis and its relevance in identifying SERPINE1-related signaling pathways in gastric cancer?

Q10-What are the potential implications and clinical significance of the enrichment of signaling pathways related to TGF-β, JAK/STAT, and Wnt in gastric cancer samples with high SERPINE1 expression?

Comments on the Quality of English Language

The quality of English language in the provided text is generally good, but there are a few areas where improvement is needed. Here are some comments:

Grammar and Syntax: Overall, the grammar and syntax are sound. However, there are a few sentences where the structure could be simplified for clarity. For example, in the sentence, "The authors also sought to explore the correlation between SERPINE1 expression and various clinicopathological features in TCGA patients," the phrasing could be streamlined for better readability.

Clarity and Conciseness: The text is informative but at times can be a bit verbose. Some sentences could be made more concise without losing essential information. For instance, "The findings were confirmed with 136 patients, that is, SERPINE1 expression was associated with poor OS [hazard ratio (HR): 1.82; 95% confidence interval (CI): 0.84-1.83; p = 0.012)] as an independent predictor (HR: 2.11, 95% CI: 0.81-2.34; p = 0.009)" could be made more concise for clarity.

Punctuation: The use of punctuation marks like parentheses, brackets, and commas is generally appropriate. However, in some cases, it could be clearer with the placement of these marks. For example, "hazard ratio (HR): 1.82; 95% confidence interval (CI): 0.84-1.83; p = 0.012" might benefit from different punctuation placement for improved readability.

Acronyms: While acronyms like TCGA are commonly used in scientific literature, it's a good practice to spell out the full form the first time it's mentioned, followed by the acronym in parentheses. This aids in understanding for readers who may not be familiar with the abbreviation.

Verb Tense: The choice of verb tenses is consistent and appropriate for a research article.

Author Response

Comments and Suggestions for Authors

The aim of this study was to investigate the association between the expression of Serpin family E member 1 (SERPINE1) and the prognosis of gastric adenocarcinoma. The study utilized data from The Cancer Genome Atlas (TCGA) to examine how SERPINE1 expression affects the survival of gastric cancer patients. Additionally, the study aimed to validate these findings by analyzing 136 samples from patients at the Affiliated Huaian No.1 People's Hospital of Nanjing Medical University using qRT-PCR and Western blot. The authors also sought to explore the correlation between SERPINE1 expression and various clinicopathological features in TCGA patients. Finally, the study aimed to uncover potential molecular mechanisms associated with SERPINE1 expression in gastric cancer through Gene Set Enrichment Analysis (GSEA). The results showed that SERPINE1 was overexpressed in tumor tissues compared to normal mucosae, and this overexpression was associated with poor overall survival of gastric cancer patients. GSEA analysis revealed that SERPINE1 overexpression was linked to the activation of cancer-related signaling pathways, including the transforming growth factor beta (TGF-β), Janus Tyrosine Kinase-Signal Transducer and Activator of Transcription (JAK/STAT), and Wnt pathways. In conclusion, the study demonstrated that SERPINE1 overexpression is associated with a poor prognosis in gastric cancer, and it provided insights into the potential molecular mechanisms underlying this association.

Q1- Can the authors explain the distribution of age among the 406 gastric cancer patients in this study and its potential implications for the research findings?

Reply: thank you for pointing out an interesting question. According to Table 1, the age distribution of 406 cancer patients ranged from 30 to 90, with an average of 65.7. Besides, we revised and added age distribution in Figure 2H; However, we did not observe significant age differences between the SERPINE1high and low gene expression groups. Therefore, we speculate that age have no effect on the expression of SERPINE1. We hope our response can meet your requirements

Table 1. The clinicopathological variables of TCGA patients with gastric cancer

Variables

Total (N=406)

%

Age at diagnosis

65.7 (30-90)

Gender

Male

256

63.1

Female

150

36.9

Histological Grade

Well-differentiated

10

2.5

Moderately differentiated

149

37.3

Poorly differentiated

240

60.2

TNM Stage

I

56

14.7

II

118

31.1

III

167

43.9

IV

39

10.3

Tumor stage

T1

23

5.8

T2

85

21.5

T3

185

46.7

T4

103

26

Node positivity

N0

122

31.4

N1

109

28

N2

80

20.6

N3

78

20

Distant metastasis

Positive

361

93

Negative

27

7

Fig. 2. Expression and association of SERPINE1 with clinicopathological characteristics in TCGA-STAD dataset.

(A) SERPINE1expression in STAD tissues and normal tissues (Wilcoxon rank sum test, ***p < 0.001). (B) SERPINE1 expression in STAD tissues and adjacent noncancerous tissues (Wilcoxon signed rank test, ***p < 0.001). (C) Expression level of SERPINE1 in patients with different pathological stages (Kruskal–Wallis test, ***p < 0.001). (D) Expression level of SERPINE1 in patients with different histologic grades (Kruskal–Wallis test, ***p < 0.001). (E,F,G) Expression level of SERPINE1 in patients with TNM (Kruskal–Wallis test, ***p < 0.001, NS, not significant). (H) Expression level of SERPINE1 in patients with age (Wilcoxon rank sum test, NS, not significant). (I) Expression level of SERPINE1 in patients with gender (Wilcoxon rank sum test, NS, not significant). (J) Overall survivals of patients with high and low SERPINE1 expression (log-rank test, p = 0.000). (K) A receiver operating characteristic (ROC) curve and the area under the curve (AUC) of SERPINE1 in 1-year, 3-year, and 5-year.

Q2- What is the significance of the observed gender distribution in this study, and does it impact the results related to SERPINE1 expression?

Reply: thank you for pointing out an interesting question. According to Table 2, We did not observe significant gender differences between the SERPINE1high and low gene expression groups. Besides, we added gender distribution in Figure 2I; However, we did not observe significant gender differences between the SERPINE1high and low gene expression groups. Therefore, we speculate that gender have no effect on the expression of SERPINE1. We hope our response can meet your requirements

Table 2. Association of SERPINE1 expression with clinicopathologic variables for TCGA patients

Variables

N

Odds ratio in SERPINE1 expression

p value

Age (continuous)

342

1.31 (0.86-2.01)

0.21

Gender (male vs. female)

344

1.02 (0.66-1.58)

0.94

Histological Grade (well vs. poorly differentiated)

337

1.84 (0.44-9.17)

0.042

TNM Stage

II vs. I

152

1.40 (1.72-2.87)

0.29

III vs. I

185

1.44 (0.73-2.72)

0.15

IV vs. I

85

1.75 (0.72-4.27)

0.064

T

T2 vs. T1

94

2.05 (0.72-6.37)

0.081

T3 vs. T1

176

2.22 (0.82-7.52)

0.042

T4 vs. T1

104

3.56 (0.91-9.86)

0.004

N (positive vs. negative)

328

1.11 (0.63-1.96)

0.72

Distant metastasis (positive vs. negative)

322

1.31 (0.64-2.72)

0.45

Fig. 2. Expression and association of SERPINE1 with clinicopathological characteristics in TCGA-STAD dataset.

(A) SERPINE1expression in STAD tissues and normal tissues (Wilcoxon rank sum test, ***p < 0.001). (B) SERPINE1 expression in STAD tissues and adjacent noncancerous tissues (Wilcoxon signed rank test, ***p < 0.001). (C) Expression level of SERPINE1 in patients with different pathological stages (Kruskal–Wallis test, ***p < 0.001). (D) Expression level of SERPINE1 in patients with different histologic grades (Kruskal–Wallis test, ***p < 0.001). (E,F,G) Expression level of SERPINE1 in patients with TNM (Kruskal–Wallis test, ***p < 0.001, NS, not significant). (H) Expression level of SERPINE1 in patients with age (Wilcoxon rank sum test, NS, not significant). (I) Expression level of SERPINE1 in patients with gender (Wilcoxon rank sum test, NS, not significant). (J) Overall survivals of patients with high and low SERPINE1 expression (log-rank test, p = 0.000). (K) A receiver operating characteristic (ROC) curve and the area under the curve (AUC) of SERPINE1 in 1-year, 3-year, and 5-year.

Q3- How do the histological grades of the tumors correlate with SERPINE1 expression, and what might this imply about gastric cancer progression?

Reply: thank you for pointing out an interesting question., we added histological grades distribution in Figure 2D; However, we did observe significant differences in SERPINE1 expression among the differential histological grades (Normal, G1, G2, G3) differences, which suggesting that the gene is involved in the progression of gastric cancer and. We hope our response can meet your requirements.

Fig. 2. Expression and association of SERPINE1 with clinicopathological characteristics in TCGA-STAD dataset.

(A) SERPINE1expression in STAD tissues and normal tissues (Wilcoxon rank sum test, ***p < 0.001). (B) SERPINE1 expression in STAD tissues and adjacent noncancerous tissues (Wilcoxon signed rank test, ***p < 0.001). (C) Expression level of SERPINE1 in patients with different pathological stages (Kruskal–Wallis test, ***p < 0.001). (D) Expression level of SERPINE1 in patients with different histologic grades (Kruskal–Wallis test, ***p < 0.001). (E,F,G) Expression level of SERPINE1 in patients with TNM (Kruskal–Wallis test, ***p < 0.001, NS, not significant). (H) Expression level of SERPINE1 in patients with age (Wilcoxon rank sum test, NS, not significant). (I) Expression level of SERPINE1 in patients with gender (Wilcoxon rank sum test, NS, not significant). (J) Overall survivals of patients with high and low SERPINE1 expression (log-rank test, p = 0.000). (K) A receiver operating characteristic (ROC) curve and the area under the curve (AUC) of SERPINE1 in 1-year, 3-year, and 5-year.

Q4- Among the TCGA patients, can the authors elaborate on the distribution of TNM stages and how they relate to SERPINE1 expression?

Reply: thank you for pointing out an interesting question., we added TNM stages distribution in Figure 2E,F,G; However, we did observe significant differences in SERPINE1 expression among the differential histological grades (T, M) differences, which suggesting that the gene is involved in the invasiveness and metastasis of gastric cancer. We hope our response can meet your requirements.

Fig. 2. Expression and association of SERPINE1 with clinicopathological characteristics in TCGA-STAD dataset.

(A) SERPINE1expression in STAD tissues and normal tissues (Wilcoxon rank sum test, ***p < 0.001). (B) SERPINE1 expression in STAD tissues and adjacent noncancerous tissues (Wilcoxon signed rank test, ***p < 0.001). (C) Expression level of SERPINE1 in patients with different pathological stages (Kruskal–Wallis test, ***p < 0.001). (D) Expression level of SERPINE1 in patients with different histologic grades (Kruskal–Wallis test, ***p < 0.001). (E,F,G) Expression level of SERPINE1 in patients with TNM (Kruskal–Wallis test, ***p < 0.001, NS, not significant). (H) Expression level of SERPINE1 in patients with age (Wilcoxon rank sum test, NS, not significant). (I) Expression level of SERPINE1 in patients with gender (Wilcoxon rank sum test, NS, not significant). (J) Overall survivals of patients with high and low SERPINE1 expression (log-rank test, p = 0.000). (K) A receiver operating characteristic (ROC) curve and the area under the curve (AUC) of SERPINE1 in 1-year, 3-year, and 5-year.

Q5- Could the authors provide further details on the relationship between T stage and SERPINE1 expression in gastric cancer patients?

Reply: thank you for pointing out an interesting question., we added T stages distribution in Figure 2E; However, we did observe significant differences in SERPINE1 expression among the differential histological grades (T, M) differences, which suggesting that the gene is involved in the invasiveness and metastasis of gastric cancer. We hope our response can meet your requirements.

Fig. 2. Expression and association of SERPINE1 with clinicopathological characteristics in TCGA-STAD dataset.

(A) SERPINE1expression in STAD tissues and normal tissues (Wilcoxon rank sum test, ***p < 0.001). (B) SERPINE1 expression in STAD tissues and adjacent noncancerous tissues (Wilcoxon signed rank test, ***p < 0.001). (C) Expression level of SERPINE1 in patients with different pathological stages (Kruskal–Wallis test, ***p < 0.001). (D) Expression level of SERPINE1 in patients with different histologic grades (Kruskal–Wallis test, ***p < 0.001). (E,F,G) Expression level of SERPINE1 in patients with TNM (Kruskal–Wallis test, ***p < 0.001, NS, not significant). (H) Expression level of SERPINE1 in patients with age (Wilcoxon rank sum test, NS, not significant). (I) Expression level of SERPINE1 in patients with gender (Wilcoxon rank sum test, NS, not significant). (J) Overall survivals of patients with high and low SERPINE1 expression (log-rank test, p = 0.000). (K) A receiver operating characteristic (ROC) curve and the area under the curve (AUC) of SERPINE1 in 1-year, 3-year, and 5-year.

Q6- What are the implications of the findings that SERPINE1 expression is higher in tumor tissues compared to normal tissues? How does this relate to the study's objectives?

Reply: thank you for pointing out an interesting question., we added the SERPINE1 expression profile in pan-cancer and normal tissue (Figure 1A-1C), and Pathological stage (Figure 2C), Histological grade (Figure 2D), TNM (Figure 2E-2G), predictive ability (Figure 2K).

Firstly, from the perspective of pan cancer, we demonstrate that SERPINE1 expression are highly expressed in the vast majority of tumors and suggest poor prognosis. Secondly, from the perspective of TNM, we found that SERPINE1 expression is closely related to the metastasis of gastric cancer (Figure 2G).

We hope our response can meet your requirements.

Fig. 1. Expression and overall survival association of SERPINE1 in Pan cancer. A Expression analysis of SERPINE1 mRNA from Pan cancer and normal tissues. B, Combining TCGA and GTEx databases to obtain SERPINE1 mRNA expression levels. C, Prognosis (overall survival) analysis of SERPINE1 in various cancers from the TCGA database using the Sangerbox website tool.

Fig. 2. Expression and association of SERPINE1 with clinicopathological characteristics in TCGA-STAD dataset.

(A) SERPINE1expression in STAD tissues and normal tissues (Wilcoxon rank sum test, ***p < 0.001). (B) SERPINE1 expression in STAD tissues and adjacent noncancerous tissues (Wilcoxon signed rank test, ***p < 0.001). (C) Expression level of SERPINE1 in patients with different pathological stages (Kruskal–Wallis test, ***p < 0.001). (D) Expression level of SERPINE1 in patients with different histologic grades (Kruskal–Wallis test, ***p < 0.001). (E,F,G) Expression level of SERPINE1 in patients with TNM (Kruskal–Wallis test, ***p < 0.001, NS, not significant). (H) Expression level of SERPINE1 in patients with age (Wilcoxon rank sum test, NS, not significant). (I) Expression level of SERPINE1 in patients with gender (Wilcoxon rank sum test, NS, not significant). (J) Overall survivals of patients with high and low SERPINE1 expression (log-rank test, p = 0.000). (K) A receiver operating characteristic (ROC) curve and the area under the curve (AUC) of SERPINE1 in 1-year, 3-year, and 5-year.

Q7- Can the authors explain how the logistic regression analysis was used to categorize gastric cancer samples into SERPINE1-high and low expression groups, and what clinical significance does this hold?

Reply: thank you for pointing out an interesting question., Based on logistic regression analysis, we separated the gastric cancer samples into SERPINE1-high and low expression groups using the median level as the cut-off value. We found that high SERPINE1 expression was associated with poor prognostic clinicopathological features, such as a poorly differentiated histological grade [odds ratio (OR) = 1.84 (0.44 - 9.17) versus (vs.) a well-differentiated grade, T3 vs. T1 T stage (OR = 2.22 (0.82 - 7.52) and T4 vs. T1 (OR = 3.56 (0.91 - 9.86; Table 2)].

Table 2. Association of SERPINE1 expression with clinicopathologic variables for TCGA patients

Variables

N

Odds ratio in SERPINE1 expression

p value

Age (continuous)

342

1.31 (0.86-2.01)

0.21

Gender (male vs. female)

344

1.02 (0.66-1.58)

0.94

Histological Grade (well vs. poorly differentiated)

337

1.84 (0.44-9.17)

0.042

TNM Stage

II vs. I

152

1.40 (1.72-2.87)

0.29

III vs. I

185

1.44 (0.73-2.72)

0.15

IV vs. I

85

1.75 (0.72-4.27)

0.064

T

T2 vs. T1

94

2.05 (0.72-6.37)

0.081

T3 vs. T1

176

2.22 (0.82-7.52)

0.042

T4 vs. T1

104

3.56 (0.91-9.86)

0.004

N (positive vs. negative)

328

1.11 (0.63-1.96)

0.72

Distant metastasis (positive vs. negative)

322

1.31 (0.64-2.72)

0.45

Q8- In the multivariate analysis, what is the significance of gender and SERPINE1 expression as predictors of survival, and how do they compare to other clinicopathological variables?

Reply: thank you for pointing out an interesting question., Univariate analysis clearly showed an inverse relationship between the poor OS and SERPINE1 expression with a hazard ratio [HR] of 1.82 and 95% confidence interval [CI] of 0.84-1.83 (p = 0.012; Table 3). Other clinicopathological variables, such as lymph node metastasis, T stage, and TNM stage, were also associated with a poor prognosis for patients (Table 3). Multivariate analysis showed that only gender and SERPINE1 expression predicted the survival of these patients, with an HR of 2.11 (95% CI: 0.81-2.34, p = 0.009; Table 3).

Q9- Could the authors provide more details on the bioinformatic analysis and its relevance in identifying SERPINE1-related signaling pathways in gastric cancer?

Reply: thank you for pointing out an interesting question., we added the association analysis between the SERPINE1 expression profile and cancer hallmark pathway (Figure 4). Besides, we inferred SERPINE1-related signaling pathways based GSEA analysis by differential analysis and guilt by association (Figure 5).

As shown in Figure 4A, we found that the most significantly hallmark pathways, including MYC_TARGETS_V1, EPITHELIAL_MESENCHYMAL_TRANSITION, OXIDATIVE_PHOSPHORYLATION, and TNFA_SIGNALING_VIA_NFKB were upregulated in gastric cancer samples with high SERPINE1 expression. Besides, EPITHELIAL_MESENCHYMAL_TRANSITION, TGF_BETA_SIGNALING, HYPOXIA, and ANGIOGENESIS were highly associated with SERPINE1 expression (Figure 4B). Specifically, ANGIOGENESIS and EPITHELIAL_MESENCHYMAL_TRANSITION remain the highest associate with SERPINE1 expression (spearman correlation coefficient >0.6, P-value < 0.05, Figure 4C-4D). Besides, we adopt two strategies to explore the correlation between SERPINE1 expression and tumor hallmarks, including differential expression analysis (Figure 5A) and guilt of association (Figure 5B). We found that the most significantly enriched signaling pathways in gastric cancer samples with high SERPINE1 expression according to the normalized enrichment score (NES) were pathways related to cancer, specifically the EPITHELIAL_MESENCHYMAL_TRANSITION, TNFA_SIGNALING_VIA_NFKB, INFLAMMATORY_RESPONSE, KRAS_SIGNALING_UP and ALLOGRAFT_REJECTION (p < 0.05, Figure 5A). Similarity, genes related to SERPINE1 expression were significantly enriched in EPITHELIAL_MESENCHYMAL_TRANSITION, TNFA_SIGNALING_VIA_NFKB, INFLAMMATORY_RESPONSE, ANGIOGENESIS, and HYPOXIA(p < 0.05, Figure 5B).

Fig. 4. Association of SERPINE1 expression and Tumor Hallmarks based on differential expression and correlation analysis. A, Differential 50 cancer hallmark score between high- and low- SERPINE1 groups. B, The correlation between SERPINE1 expression and 50 cancer hallmark score. C-D, The most prominent cancer hallmark associated with differential co-expression of SERPINE1 expression (spearman correlation coefficient >0.6, P-value < 0.05).

Fig. 5. Identification of SERPINE1-associated gene pathways using gene set enrichment analysis (GSEA). A, Gene set enrichment analysis for patients classified into SERPINE1 high group and SERPINE1 low group according to the median value of SERPINE1 expression. B, Via a guilt by association approach, we performed GSEA to explore the potential cancer hallmark association of SERPINE1 in stomach cancer. NES, normalized enrichment score; NOM-p value, nominal p value.

Q10-What are the potential implications and clinical significance of the enrichment of signaling pathways related to TGF-β, JAK/STAT, and Wnt in gastric cancer samples with high SERPINE1 expression?

Reply: thank you for pointing out an interesting question., we have removed the irrelevant content in this section and replaced it with Figure 5

Comments on the Quality of English Language

The quality of English language in the provided text is generally good, but there are a few areas where improvement is needed. Here are some comments:

Grammar and Syntax: Overall, the grammar and syntax are sound. However, there are a few sentences where the structure could be simplified for clarity. For example, in the sentence, "The authors also sought to explore the correlation between SERPINE1 expression and various clinicopathological features in TCGA patients," the phrasing could be streamlined for better readability.

Reply: thank you for pointing out a question. We have modified the content of the corresponding section.

Clarity and Conciseness: The text is informative but at times can be a bit verbose. Some sentences could be made more concise without losing essential information. For instance, "The findings were confirmed with 136 patients, that is, SERPINE1 expression was associated with poor OS [hazard ratio (HR): 1.82; 95% confidence interval (CI): 0.84-1.83; p = 0.012)] as an independent predictor (HR: 2.11, 95% CI: 0.81-2.34; p = 0.009)" could be made more concise for clarity.

Reply: thank you for pointing out a question. We have revised the content of the corresponding section.

Punctuation: The use of punctuation marks like parentheses, brackets, and commas is generally appropriate. However, in some cases, it could be clearer with the placement of these marks. For example, "hazard ratio (HR): 1.82; 95% confidence interval (CI): 0.84-1.83; p = 0.012" might benefit from different punctuation placement for improved readability.

Reply: thank you for pointing out a question. We have revised the content of the corresponding section.

Acronyms: While acronyms like TCGA are commonly used in scientific literature, it's a good practice to spell out the full form the first time it's mentioned, followed by the acronym in parentheses. This aids in understanding for readers who may not be familiar with the abbreviation.

Reply: thank you for pointing out a question. We have revised the content of the corresponding section. TGCA first appeared in the document as The Cancer Genome Atlas (TCGA)

Verb Tense: The choice of verb tenses is consistent and appropriate for a research article.

Reply: thank you for pointing out a question. We have revised the content of the corresponding section.

Reviewer 2 Report

Comments and Suggestions for Authors

This is interesting work, providing information related to the possibility of early diagnosis of cancer, an approach that can be considered for implementation in clinical practice.

The paper is interesting, and in my opinion deserve to be considered for publication. However, some adjustments will need to be taken in considerations by the authors of the manuscript.

Ln5,6: Please, affiliations for the authors will need to be with complete addresses.

Maybe in the abstract will be appropriate to remove words "Background", "methods", etc. And do it as one single text without sections.

Ln46: Can you provide reference regarding the effect of consumption of pickled vegetables as pro-cancer factor, since other works showing that these food products can be source of beneficial microbes. In this case, a reference will be more than appropriate. 

Ln54: please, remove the hyperlink from the web-page citation.

Ln57: Please, be more specific and provide exact year instead of "recent tears" term.

Ln93, 96, and other similar cases - please, use italics for the "in vitro", "in vivo", etc.

Ln130: In this and similar cases, please, use address for the headquarter and not for the local distributors for used material or equipment.

In the previous text was mentioned 136 cases. What is correct? Please, explain and if needed, correct.

Please, references need to be formatted according to the journal instructions.

Author Response

Comments and Suggestions for Authors

This is interesting work, providing information related to the possibility of early diagnosis of cancer, an approach that can be considered for implementation in clinical practice.

The paper is interesting, and in my opinion deserve to be considered for publication. However, some adjustments will need to be taken in considerations by the authors of the manuscript.

Ln5,6: Please, affiliations for the authors will need to be with complete addresses.

Reply: thank you for pointing out a question. We have added a complete addresses for affiliations, as following Department of Radiotherapy, the Affiliated Huaian No.1 People’s Hospital of Nanjing Medical University, No.1, Huanghe West Road, Huaiyin District, Huai'an City, Jiangsu Province. 223300, China.

Maybe in the abstract will be appropriate to remove words "Background", "methods", etc. And do it as one single text without sections.

Reply: thank you for pointing out a question. We revised as following,

Gastric cancer, is a common malignancy with a heavy burden, and is still a significant global health problem. Aberrant expression of Serpin family E member 1 (SERPINE1) is associated with multiple cancer progression. We used to 406 The Cancer Genome Atlas (TCGA) STAD dataset and 136 in house gastric cancer dataset investigate alteration of SERPINE1 expression for association with gastric adenocarcinoma prognosis. Subsequently, qRT-PCR and Western blot were used to validate the expression level of SERPINE1 between tumor and adjacent normal tissues. The correlation between the expression of SERPINE1 with clinicopathological features in TCGA patients was analyzed using Wilcoxon signed-rank and logistic regression tests. The potential molecular mechanism associated with SERPINE1 expression in gastric cancer were confirmed using Gene Set Enrichment Analysis (GSEA). The TCGA data showed that SERPINE1 was overexpressed in tumor tissues compared to that of normal mucosae and associated with the tumor T stage and pathological grade. SERPINE1 overexpression was associated with poor overall survival (OS) of patients. The findings were confirmed with 136 patients, that is, SERPINE1 expression was associated with poor OS [hazard ratio (HR): 1.82; 95% confidence interval (CI): 0.84-1.83; p = 0.012)] as an independent predictor (HR: 2.11, 95% CI: 0.81-2.34; p = 0.009). The GSEA data showed that SERPINE1 overexpression was associated with activation of the EPITHELIAL_MESENCHYMAL_TRANSITION, TNFA_SIGNALING_VIA_NFKB, INFLAMMATORY_RESPONSE, ANGIOGENESIS, and HYPOXI in cancer.

SERPINE1 overexpression is associated with poor gastric cancer prognosis as well as activation of the pathways related to cancer such as the TGF-β, JAK/STAT, and Wnt signaling pathways.

Ln46: Can you provide reference regarding the effect of consumption of pickled vegetables as pro-cancer factor, since other works showing that these food products can be source of beneficial microbes. In this case, a reference will be more than appropriate.

Reply: thank you for pointing out a question. This is a very good proposal.

Ln54: please, remove the hyperlink from the web-page citation.

Reply: thank you for pointing out a question. We deleted the hyperlink.

Ln57: Please, be more specific and provide exact year instead of "recent tears" term.

Reply: thank you for pointing out a question. We have revised this section of the content.

Ln93, 96, and other similar cases - please, use italics for the "in vitro", "in vivo", etc.

Reply: thank you for pointing out a question. We have revised this section of the content.

Ln130: In this and similar cases, please, use address for the headquarter and not for the local distributors for used material or equipment.

Reply: thank you for pointing out a question. We have revised this section of the content.

In the previous text was mentioned 136 cases. What is correct? Please, explain and if needed, correct.

Reply: thank you for pointing out a question. We have revised this section of the content

Please, references need to be formatted according to the journal instructions.

Reply: thank you for pointing out a question. We have revised the reference format according to the journal guidance.

Reviewer 3 Report

Comments and Suggestions for Authors

The paper is interesting, well-designed, and well-written, though there is very limited touch with mechanisms. Here are some minor comments:

1. Fig. 1: the legend should be extended and revised accordingly

2. Fig. 2: it's a little bit confusing for Fig.2c and d with similar figures. I recommend adding some annotations in the figure to clarify

3. Discussion: Since there are limited mechanisms explored in the paper, innate immunity (PMID: 27706894) can also be involved in potential discusson.

Comments on the Quality of English Language

Minor editing of the English language required

Author Response

Comments and Suggestions for Authors

The paper is interesting, well-designed, and well-written, though there is very limited touch with mechanisms. Here are some minor comments:

  1. Fig. 1: the legend should be extended and revised accordingly

Reply: thank you for pointing out an interesting question. Based on the comments of other reviewers, we revised the figure1 and figure2 as follows,

Fig. 1. Expression and overall survival association of SERPINE1 in Pan cancer. A Expression analysis of SERPINE1 mRNA from Pan cancer and normal tissues. B, Combining TCGA and GTEx databases to obtain SERPINE1 mRNA expression levels. C, Prognosis (overall survival) analysis of SERPINE1 in various cancers from the TCGA database using the Sangerbox website tool.

Fig. 2. Expression and association of SERPINE1 with clinicopathological characteristics in TCGA-STAD dataset.

(A) SERPINE1expression in STAD tissues and normal tissues (Wilcoxon rank sum test, ***p < 0.001). (B) SERPINE1 expression in STAD tissues and adjacent noncancerous tissues (Wilcoxon signed rank test, ***p < 0.001). (C) Expression level of SERPINE1 in patients with different pathological stages (Kruskal–Wallis test, ***p < 0.001). (D) Expression level of SERPINE1 in patients with different histologic grades (Kruskal–Wallis test, ***p < 0.001). (E,F,G) Expression level of SERPINE1 in patients with TNM (Kruskal–Wallis test, ***p < 0.001, NS, not significant). (H) Expression level of SERPINE1 in patients with age (Wilcoxon rank sum test, NS, not significant). (I) Expression level of SERPINE1 in patients with gender (Wilcoxon rank sum test, NS, not significant). (J) Overall survivals of patients with high and low SERPINE1 expression (log-rank test, p = 0.000). (K) A receiver operating characteristic (ROC) curve and the area under the curve (AUC) of SERPINE1 in 1-year, 3-year, and 5-year.

  1. Fig. 2: it's a little bit confusing for Fig.2c and d with similar figures. I recommend adding some annotations in the figure to clarify

Reply: thank you for pointing out an interesting question. We revised figure 2 as follow,

Fig. 2. Expression and association of SERPINE1 with clinicopathological characteristics in TCGA-STAD dataset.

(A) SERPINE1expression in STAD tissues and normal tissues (Wilcoxon rank sum test, ***p < 0.001). (B) SERPINE1 expression in STAD tissues and adjacent noncancerous tissues (Wilcoxon signed rank test, ***p < 0.001). (C) Expression level of SERPINE1 in patients with different pathological stages (Kruskal–Wallis test, ***p < 0.001). (D) Expression level of SERPINE1 in patients with different histologic grades (Kruskal–Wallis test, ***p < 0.001). (E,F,G) Expression level of SERPINE1 in patients with TNM (Kruskal–Wallis test, ***p < 0.001, NS, not significant). (H) Expression level of SERPINE1 in patients with age (Wilcoxon rank sum test, NS, not significant). (I) Expression level of SERPINE1 in patients with gender (Wilcoxon rank sum test, NS, not significant). (J) Overall survivals of patients with high and low SERPINE1 expression (log-rank test, p = 0.000). (K) A receiver operating characteristic (ROC) curve and the area under the curve (AUC) of SERPINE1 in 1-year, 3-year, and 5-year.

  1. Discussion: Since there are limited mechanisms explored in the paper, innate immunity (PMID: 27706894) can also be involved in potential discusson.

Reply: thank you for pointing out an interesting question. We did not mention the relationship between SERPINE1 and immunity in the manuscript. If necessary, we will explore the relationship between this gene and immunity in the next revision according to the PMID: 27706894.

Reviewer 4 Report

Comments and Suggestions for Authors

Dear authors,

In this study, the authors tried to evaluate whether SERPINE1 was linked to poor prognosis of gastric cancer. Unfortunately, highly similar study had been reported in 2022 (refer to 10.1155/2022/2647825 ). Hense, the novelty of this study was diminished.

Comments on the Quality of English Language

The quality of English writing is fine to me.

Author Response

Comments and Suggestions for Authors

Dear authors,

In this study, the authors tried to evaluate whether SERPINE1 was linked to poor prognosis of gastric cancer. Unfortunately, highly similar study had been reported in 2022 (refer to 10.1155/2022/2647825 ). Hense, the novelty of this study was diminished.

Reply: thank you for pointing out an interesting question. We have noticed some similarities between our research and previous studies (10.1155/2022/2647825). However, our research advantage lies in collecting 136 clinical samples and exploring the expression profile and clinical characteristics of SERPINE1 at the tissue level. In addition, we applied various bioinformatics methods to deeply explore the biological phenotype of SERPINE1 involved, and clarified that the EMT process is an important regulatory part of SERPINE1.

Round 2

Reviewer 3 Report

Comments and Suggestions for Authors

3. Discussion: Since there are limited mechanisms explored in the paper, innate immunity (PMID: 27706894) can also be involved in potential discusson.

Reply: thank you for pointing out an interesting question. We did not mention the relationship between SERPINE1 and immunity in the manuscript. If necessary, we will explore the relationship between this gene and immunity in the next revision according to the PMID: 27706894.

Reply: Yes, please go ahead with the discussion.

Comments on the Quality of English Language

Fine

Author Response

Response:Thank you for pointing out an interesting question. We have modified the discussion section and added the following immunological associations.

Significantly, from the SERPINE1 related GSEA, it can be seen that TNFA_SIGNALING_VIA_NFKB and INFLAMMATORY_RESPONSE is significantly related to SERPINE1, which implied that SERPINE1 is related to the immune signaling pathway. In addition, some published articles have reported a significant correlation between SERPINE1 and immunity factor in Pan-cancer[42]. cGAS-cGAMP-STING, the three musketeers of cytosolic DNA sensing and signaling[43] PMID: 27706894, initializes the immune process. In future research, we will explore the relationship between SERPINE1 and immune recognition, further elucidating the relationship between the SERPINE1's immune promotion and inhibition in gastric cancer.

Reviewer 4 Report

Comments and Suggestions for Authors

I have no further question about this study.

Author Response

Comments and Suggestions for Authors

I have no further question about this study.

Response : Thank you very much for taking the time to review this manuscript.